# RNA Interference by Ingested Dsrna-Expressing Bacteria to Study Porphyrin Pigmentation in *Crassostrea gigas*

**DOI:** 10.3390/ijms22116120

**Published:** 2021-06-06

**Authors:** Biyang Hu, Qi Li, Hong Yu

**Affiliations:** 1Key Laboratory of Mariculture, Ministry of Education, Ocean University of China, Qingdao 266003, China; hubiyang828@163.com (B.H.); hongyu@ouc.edu.cn (H.Y.); 2Laboratory for Marine Fisheries Science and Food Production Processes, Qingdao National Laboratory for Marine Science and Technology, Qingdao 266237, China

**Keywords:** shell color, pigmentation, porphyrin, RNA interference, *Crassostrea gigas*

## Abstract

Porphyrins are a widespread group of pigments in nature which are believed to contribute to shell colors in mollusks. Previous studies have provided candidate genes for porphyrin shell coloration, however, the linkage analysis between functional genes and porphyrin pigmentation remains unclear in mollusks. RNA interference is a powerful molecular tool for analyzing the loss of functions of genes in vivo and alter gene expression. In this study, we used unicellular alga *Platymonas subcordiformis* and *Nitzschia closterium f. minutissima* as vectors to feed oysters with *Escherichia coli* strain HT115 engineered to express double-stranded RNAs targeting specific genes involved in porphyrin synthesis. A strain of *Crassostrea gigas* with orange shell was used to target key haem pathway genes expression using the aforementioned approach. We show here that feeding the oysters with *E. coli*, containing dsRNA targeting pigmentation genes, can cause changes in the color of the newly deposited shell. For example, the RNAi knockdown of *CgALAS* and *CgPBGD* resulted in the loss of uroporphyrin pigmentation from the shell due to the accumulation of the pigment in the oyster’s mantle. The study probed the crucial role of ALAS and PBGD genes potential functions of uroporphyrin production and shell color pigmentation in *C. gigas*.

## 1. Introduction

The diversity of mollusks shell colors has been well reported [1,2,3,4]. Pigments deposited by mantle determined the shell color [5]. The outer fold of the mantle contributes to shell formation [6]. The dorsal mantle epithelium secretes the mollusk’s shell and controls pigmentation [7,8]. The formation of shell color has been studied in mollusks such as marine snails, Pacific oyster, pearl mussel and black-lipped pearl oyster [9,10,11,12]. Moreover, several lncRNA and mRNA transcripts associated with shell pigmentation have been identified which influence pigment biosynthesis including melanin, carotenoid and tetrapyrrole [13].

Porphyrins are a widespread group of pigments in nature which are believed to contribute to shell colors in mollusks [14], often leading to red, brown or purple shell pigmentation [3]. Among porphyrins, the uroporphyrin I and III are found in mollusks [15,16,17,18]. It has been suggested that porphyrin in shell pigmentation is produced de novo by the animal through the haem pathway, the production of uroporphyrin is derived from the oxidation of uroporphyrinogen I and III which belong to non-enzymatic side paths [17,19]. The regulation of the haem pathway initially occurs through the rate-limiting enzyme aminole-vulinic acid synthase (ALAS), by downregulating its transcription, upregulating mRNA breakdown, blocking its uptake into mitochondria, and increasing the breakdown of the protein in mammals [20]. The second gene in the haem pathway aminole-vulinic acid dehydratase (ALAD) has been proven to catalyze the formation of porphobilinogen by the dimerization of two molecules of 5-aminolaevulinic acid (ALA). Then, the porphobilinogen deaminase (PBGD) condenses two molecules of porphobilinogen to form hydroxymethylbilane (HMB). The fourth enzyme uroporphyrinogen III synthase (UROS) subsequently converts HMB into uroporphyrinogen III by closing the linear tetrapyrrole molecule to form a ring (Ajioka et al., 2006), followed by the activation of the fifth enzyme uroporphyrinogen decarboxylase (UROD) which builds up the uroporphyrinogen III. Uroporphyrin formation can be enhanced when the metabolites ALA, porphobilinogen, and HMB are overproduced [21]. Furthermore, increasing levels of ALAS, along with decreasing levels of UROS and UROD, could simultaneously increase the production of uroporphyrin [22]. Genes in the haem pathway associated with the production of porphyrin pigments in tissues have been identified in marine snail *Clanculus margaritarius* and *C. pharaonius* [9]. In addition, a recent study indicated that 5-aminolaevulinate synthase (ALAS), which is the first and rate-limiting enzyme in this pathway plays an important role in shell coloration in Yesso scallops *Patinopecten yessoensis* [23]. The activation of ALAS and PBGD were indicated to own functions of increasing the uroporphyrin production, and UROS was claimed to decrease the level of uroporphyrin I. These studies provide candidate genes for porphyrin shell coloration. However, the linkage analysis between these functional genes and porphyrin pigmentation remains unclear in mollusks.

RNA interference (RNAi) is a powerful molecular tool for analyzing the loss of functions of genes in vivo and alter gene expression. RNAi can be delivered in a variety of ways, for example soaking animals in dsRNA or morpholinos, injecting dsRNA or morpholinos, or feeding dsRNA-expressing bacteria [24]. The traditional injection method has been described in mollusks, but this method is not only technically challenging but also harmful to the regular growth of organisms. A recent study successfully downregulated the tyrosinase gene of the Manila clam *Ruditapes philippinarum* by injecting dsRNA into Manila clam, however, there was no obvious change in the shell-color phenotypes indicating that the short-term knockdown of shell-color related genes may not rapidly change the phenotype of mollusks [25]. To verify the functional genes of shell color, long-term and noninvasive approaches are crucial. An effective feeding-based RNAi method was reported in the Pacific oyster *Crassostrea gigas* by using the dinoflagellate *Heterocapsa triquetra* as a vector for the dsRNA-producing *E. coli* HT115 [26]. The algae–bacteria extracellular associations were utilized to retain the algae–bacteria particles by oyster gill filtration [27,28,29,30]. Evidence for the vital role of tyrosinase in shell pigmentation of *C. gigas* was gained through this non-invasive RNAi method [31]. The algae–bacteria interaction strategy not only provides an inexpensive and high-output RNAi method, but also supports the long-term RNA interference requirements for shell-color-related gene function analysis by producing large quantities of dsRNA [32,33].

*C. gigas* is a widely distributed and economically important species in the world. It has become a potential model for mollusk-related research [34]. Although the complete *C. gigas* genome sequence is available (PRJEB35351), due to the lack of molecular techniques to manipulate gene expression, the molecular mechanism of oyster porphyrin pigmentation remains unclear. Studies in purple-shell oysters revealed the presence of uroporphyrin in the shell and mantle of *C. gigas* [18]. Through selective breeding, mutant oysters possessing pure orange shells without any other pigmentation was obtained [35]. In our previous study, porphyrins were observed on the orange-shell surface and mantles (Hu et al., unpublished data). To assess the functions of candidate genes on orange shell color formation, five genes (*CgALAS*, *CgALAD*, *CgPBGD*, *CgUROS*, *CgUROD*) encoding haem pathway enzymes that are potentially critical for uroporphyrinogens synthesis were analyzed through the feeding-based RNAi experiment. The interference efficiency was verified through qPCR. Uroporphyrin pigmentation was characterized in newly deposited shells. Candidate genes were successfully knocked down which led to the pigmentation phenotype changes in *C. gigas*. The study demonstrated the affection of downregulating key haem pathway genes in shell color formation and probed the crucial role of ALAS, PBGD and UROS genes potential functions of uroporphyrin production and pigmentation.

## 2. Results

### 2.1. Plasmid Constructs and Induction Assays

The expression of dsRNAs was induced in *E. coli* strains which were, respectively, engineered with 0.4 mM IPTG for 4 h at 37 °C. Total RNA was isolated from the induced and non-induced bacteria and analyzed via 1% agarose gel electrophoresis. Bands of dsRNA corresponding to the ALAS, ALAD, PBGD, UROS, UROD and EGFP genes plus sequence between two T7 promoters of about 100 bp was observed in the IPTG-induced *E. coli* transformed with *CgALAS*, *CgALAD*, *CgPBGD*, *CgUROS*, *CgUROD* and EGFP construct (Figure 1), but not in the non-induced *E. coli* transformed with *CgALAS* (Figure 1a), *CgALAD* (Figure 1b), *CgPBGD* (Figure 1c), *CgUROS* (Figure 1d), *CgUROD* (Figure 1e) and EGFP (Figure 1f) construct, indicating the corresponding expression of dsRNA in *E. coli*.

### 2.2. Impact of RNAi on Phenotypic Change

Oysters were fed with alga mixed with six different types of *E. coli* culture expressing distinct types of dsRNAs. Oysters were classified as growing obviously newly developed shell or not (Table 1 and Figure 2). Seven, nine and eight of the oysters fed on bacteria-expressing *CgALAS*, *CgALAD* and *CgPBGD* dsRNA displayed the newly deposited shell growth (Table 1). At the same time, the newly deposited shell growth was detected in nine, ten and ten oysters from the *CgUROS*, *CgUROD* and EGFP groups, respectively.

Compared with the images at the beginning of the experiment, the newly deposited shells of the ALAS and PBGD group represented a remarkable loss of orange pigmentation than the EGFP group (Figure 2). The *L*a*b** value of the newly deposited shells was calculated to show the variations of the orange shell color phenotype (Figure 3). Oysters fed on *E. coli*-expressing dsRNA EGFP showed no significant changes in orange shell pigmentation (Figure 2c). In contrast, a number of oysters in the experiment groups displayed the orange pigmentation gain or loss phenotype. According to the *L*a*b** value analysis, the *a** and *b** value of the ALAS group was significantly lower than the control group (Figure 3). As for the ALAD group, there were no significant phenotype changes in the shell orange pigmentation, and there was no significant difference in the *L*a*b** value compared with the control group. It is noteworthy that oysters fed on the PBGD dsRNA displayed an obvious orange pigmentation loss. The newly deposited shell of PBGD group has higher lightness value (*L**) and greater orange pigmentation loss than the control group (*a*b** value of PBGD group was significantly lower than the control group). In contrast, the UROS UROD group showed more visible orange pigmentation and higher *a*b** value than the EGFP group but was not significant.

### 2.3. Detection of RNAi Product in Mantle Tissues

The effectiveness of dsRNA delivery to oyster mantle tissues by feeding was determined by analyzing dsRNA in oysters fed on bacteria containing the L4440-EGFP by qPCR. EGFP dsRNA expression was detected in oysters fed on bacteria containing the L4440-EGFP plasmid but not in oysters fed on *CgALAS*-L4440, *CgALAD*-L4440, *CgPBGD*-L4440, *CgUROS*-L4440, *CgUROD*-L4440. The detection of RNAi products (*CgALAS*, *CgALAD*, *CgPBGD*, *CgUROS*, *CgUROD* dsRNA) in oyster mantles was assessed as a function of the ratio of the interfering *CgALAS*, *CgALAD*, *CgPBGD*, *CgUROS*, *CgUROD* dsRNAs to the endogenous *CgALAS*, *CgALAD*, *CgPBGD*, *CgUROS*, *CgUROD* mRNA levels (Figure 4). A significant amount of RNAi products was found in *CgALAS* and *CgUROS* dsRNA oysters compared to EGFP dsRNA oysters at day 15 and at day 30. Significant amount of RNAi products was also found in *CgALAD*, *CgPBGD* and *CgUROD* dsRNA oysters at day 15. This is consistent with the dsRNA entrance in mantle tissues during the phase of interference treatment. At day 15, the levels of dsRNAs were 2.3, 2.1, 2.1, 3.9 and 2.1 times more abundant than endogenously expressed mRNAs in the *CgALAS*, *CgALAD*, *CgPBGD*, *CgUROS*, *CgUROD* dsRNA oysters, respectively. At day 30, the levels of dsRNAs were 2.1 and 2.3 times more abundant than endogenously expressed mRNAs in RNAi-treated oysters expressing the *CgALAS* dsRNA and *CgUROS* dsRNA, respectively. Probabilities for two-tailed, paired *t* tests for differences in gene expression levels among control and experiment groups of *C. gigas* are shown in Appendix A. 

### 2.4. Efficiency of RNAi on the Target Gene Transcription

qPCR results showed that in oysters fed on bacteria containing *CgALAS*-L4440, *CgALAD*-L4440, *CgPBGD*-L4440, *CgUROS*-L4440, *CgUROD*-L4440, dsRNA constructs showed reduced *CgALAS*, *CgALAD*, *CgPBGD*, *CgUROS*, *CgUROD* expressions compared to the control group. As shown in Figure 5, *CgALAS*, *CgPBGD* and *CgUROD* mRNA levels were significantly diminished in *C. gigas* fed on bacteria-expressing *CgALAS*, *CgPBGD* and *CgUROD* dsRNA as compared to control group at day 15. In the ALAS group, these results were significant and the relative *CgALAS* mRNA expression levels dropped down to ~62% of the control values. Thus, the average knockdown was ~38%, with a maximum knockdown value of ~37.5% (*p* = 0.0476) at day 15. A significant decrease (97%) was observed in *CgPBGD* transcripts at day 15 compared to the EGFP dsRNA oysters (*p* < 0.0001). Remarkable decrease in *CgUROD* mRNA expression levels dropped down to ~98% of the control values (*p* < 0.0001). There was no significant difference in *CgALAD/CgUROS* transcripts neither between the ALAD group or UROS group and the control at both the middle and the end of interference exposure. At day 15, individual analysis revealed that 80% of individuals in the PBGD group, and all the individuals in UROD group showed decreased gene expression by more than 70% compared to the control (Appendix A). At day 30, no individuals in experiment groups displayed decreased gene expression by more than 70% compared to the control group (Appendix A).

## 3. Discussion

Oysters are filter-feeding animals, and their open vascular system makes traditional injection RNA interference methods unable to efficiently interfere the expression of genes. The bacteria-feeding RNAi method used in this study revealed the function of genes related to uroporphyrin pigmentation in *C. gigas*. The effect of porphyrin pigments on shell color has been reported in mollusks, and uroporphyrin I and III in oyster shells have also been identified [3,17,18]. Genes associated with the uroporphyrinogen synthesis have been characterized in marine snails and pacific oysters [9]. The first five genes were identified from *C. gigas* genome and were revealed to have significant expression levels among different shell-color oysters [36]. Previous studies provide candidate genes for porphyrin pigmentation in mollusks. The effect of dsRNA interference usually reached the maximum level at 15 to 30 days after the post-induction of dsRNA in *C. gigas* [26,37,38]. According to previous studies on *C. gigas*, feeding oysters with dsRNA for 35 days conducted the successful knockdown of the corresponding genes and observed oysters with newly deposited shell growth [31]. A total 30-day knockdown experiment was designed to obtain the maximum RNAi effect and RNAi phenotype of the newly deposited shell [26,31].

The data in this study revealed that the knockdown of the five haem pathway genes has different effects on orange pigmentation. To compare the level of color change in the newly deposited shell, the color parameters of *L** (brightness), *a** (red) and *b** (yellow) between the experimental group and the control group were obtained. Results showed that the *L** of the newly deposited shell of *C. gigas* in the *CgPBGD* group was significantly higher than that of the EGFP group, which means the newly formed shell was whiter. At the same time, the *a** and *b** values of the *CgALAS* group and the *CgPBGD* group were significantly lower than those of the control group, which means that the newly deposited shells in the experimental groups had less red and yellow. Through the RNAi bacterial feeding process, compared with the control group, on the fifteenth day, the average level of transcripts in the *CgALAS* and *CgPBGD* groups significantly decreased by 38% and 97%, which indicates that the loss of orange pigmentation has remarkable relations with the decreasing transcription of *CgALAS* and *CgPBGD*. In contrast, the *CgUROS* group newly deposited shells had a higher *a** and *b** value than the control group. It is noteworthy that the average level of transcripts in the *CgUROS* group was significantly reduced by 98% compared to the control group on the 15th day. ALAS is the rate-limiting enzyme of the haem pathway [39]. Studies have shown that the increasing expression levels of ALAS could simultaneously increase the production of uroporphyrin [22]. The rising activity of PBGD has been suggested to be associated with uroporphyrin I production [40,41]. Whiter shell appears when the activity of *CgALAS* and *CgPBGD* is reduced, and the non-enzymatic side paths fail to generate the uroporphyrins at regular bases. This phenotypic change revealed the functions of *CgALAS* and *CgPBGD* on uroporphyrin pigments synthesis. Accordingly, it could be proposed that the downregulation of *CgALAS* and *CgPBGD* gene affects the accumulation of uroporphyrin pigments in mantles and affects the coloration of the newly deposited shell.

The RNAi response was highly variable among individuals in this study. It had been inferred that the change in the amount of dsRNA that eventually enters the mantle cell may largely contribute to the variability of the RNAi response [37,38]. It should be noted that the expression level of *CgALAD*, *CgPBGD*, *CgUROS* and *CgUROD* mRNA on the 30th day was lower than that on the 15th day. However, several individuals on the 30th day showed obvious high expression, such as No. 5 in the *CgUROD* group. These expression levels, which are significantly different from other individuals, affect the overall expression trend. The irregular expression level was also detected in the *CgTyr* knockdown and it has been speculated that the long-term feeding of bacteria may affect individual health and normal gene expression [31]. Although the significant downregulation of target genes was observed in this study, there are still parameters that need to be optimized. For example, the concentration of daily algae/bacterial co-inoculum, treatment duration and total number of individuals need to be optimized [26]. At the same time, when conducting research on shell-related interference, the selection of *C. gigas* at proper circadian rhythms is crucial. Higher shell growth could be obtained at water temperature between 18 °C and 24 °C, and shell development is increased in summer and autumn. Furthermore, the phenotypic change could take place in oyster mantles. With the expression modulation of the haem pathway genes, the accumulation of uroporphyrins could be affected. Therefore, the concentration of uroporphyrin between the experiment and control group could be analyzed by HPLC to make the data more detailed [17].

## 4. Materials and Methods

### 4.1. Oyster and Algal Culture

One-year-old Pacific oysters with the whole orange shell (120 individuals) were collected from the culture population in Weihai, Shandong, China (Oct 2020). Oysters were randomly stuck with labels, divided into six groups of 20 oysters, and placed into six tanks of 40 L aquarium equipped with an open circulatory system of natural seawater (30 psu) at a flow rate of 12 mL min^−1^. They were acclimated 7 days before the start of the experiment and continuously fed with *Platymonas subcordiformis* (25,000 cells mL^−1^) and *Nitzschia closterium f. minutissima* (35,000 cells mL^−1^). Alga *P. subcordiformis* and *N. closterium f. minutissima* cultures were grown in f/2 medium using sterilized seawater. Cultures were maintained at 19 ± 1 °C, under a light:dark (L:D) 14:10 cycle. *P. subcordiformis* has four flagellates and is 16–30 μm long; *N. closterium f. minutissima* has no flagellate and is 12–23 μm long.

### 4.2. Vector Construction and Expression of dsRNA

The L4440 vector was selected to generating dsRNAs in an inducible manner in *E. coli*. L4440 contains two convergent T7 polymerase promoters in opposite orientations separated by a multi-cloning site [33]. The first five genes of the haem pathway (aminole-vulinic acid synthase-*CgALAS*, aminole-vulinic acid dehydratase-*CgALAD*, porphobilinogen deaminase-*CgPBGD*, uroporphyrinogen-III synthase-*CgUROS*, uroporphyrinogen decarboxylase-*CgUROD*) were targeted for degradation by RNAi. The sequence of five genes were amplified by PCR based on the sequence information downloaded from the National Centre for Biotechnology Information (Date: 1 February 2021; NCBI, http://www.ncbi.nlm.nih.gov; Accession: PRJEB35351; ALAS: LOC105330021; ALAD: LOC105330983; PBGD: LOC105326792 UROS: LOC105336067; UROD: LOC105347428). DNA fragments of *CgALAS*, *CgALAD*, *CgPBGD*, *CgUROS*, *CgUROD* were amplified from *C. gigas* by PCR using specific primers (Table 2). Appropriate restriction enzyme sites were engineered in the 5′ ends of both PCR primers for sub-cloning. Enhanced green fluorescent protein (EGFP) was used as a negative control in the experiments as it did not have a natural target RNA in *C. gigas*. Fragments were purified and ligated into L4440 (VECT75473, Huayueyang Biotech Co., Ltd., Beijing, China) vectors, which were used to transform DH5α bacteria (Vazyme, Nanjing, China). Noted for each construct, the bacteria were cultured in Luria broth (LB) containing 100 μg mL^−1^ ampicillin. L4440 (VECT75473, Huayueyang Biotech Co., Ltd., Beijing, China), generating constructed vectors *CgALAS*-L4440, *CgALAD*-L4440, *CgPBGD*-L4440, *CgUROS*-L4440, *CgUROD*-L4440, EGFP-L4440, as confirmed by sequencing. Competent *E. coli* cells HT115 (F-*mcr*A *mcr*B, IN(*rrn*D-*rrn*E)1 *rnc*14::Tn10(DE3 lavUV5::T7 polymerase)), lacking the dsRNA-specific endonuclease RNase III, were used to produce a high level of specific dsRNAs [33]. HT115 (DE3) competent cells were prepared using standard CaCl_2_ methodology and were transformed with *CgALAS*-L4440, *CgALAD*-L4440, *CgPBGD*-L4440, *CgUROS*-L4440, *CgUROD*-L4440, *EGFP*-L4440, respectively. RNA extraction was performed on non-induced *E. coli* and IPTG-induced *E. coli* transformed with a specific gene using Bacteria RNA Extraction Kit (Vazyme, Nanjing, China).

### 4.3. RNAi Feeding Procedures

During the interference phase, six tanks of orange shell oysters were continuously exposed to the alga/dsRNA-producing bacteria co-inoculum. Six dsRNA-producing bacteria were prepared by inducing *E. coli* strain HT115 (DE3) bacteria transformed with three constructed plasmids (*CgALAS*-L4440, *CgALAD*-L4440, *CgPBGD*-L4440, *CgUROS*-L4440, *CgUROD*-L4440, *EGFP*-L4440) with isopropylβ-D-thiogalactoside (IPTG). Algae/bacteria co-inoculum was produced by mixing algal culture and bacterial suspension at a ratio of 100 bacteria per algal cell, with a final *p. subcordiformis* concentration of 25,000 cells mL^−1^ and *N. closterium f. minutissima* concentration of 35,000 cells mL^−1^. The ratio was optimized by increasing algal concentration according to Payton et al. [26]. Food reserves were renewed with fresh algae/bacteria co-inoculum every 24 h. Specifically, *E. coli* strain HT115 bacteria containing recombinant plasmid were grown overnight with shaking in LB with ampicillin (50 μg mL^−1^) and tetracycline (12.5 μg mL^−1^) at 37 °C. Five milliliters of overnight culture was diluted 100-fold in 500 mL of fresh LB medium containing ampicillin (50 μg mL^−1^) and tetracycline (12.5 μg mL^−1^) and allowed to grow to OD_595_ = 0.4. dsRNA production was induced by 0.4 mM IPTG for 4 h at 37 °C under agitation. Then, induced bacterial cultures were centrifuged and bacterial pellets were resuspended in 500 mL *P. subcordiformis* and 125 mL *N closterium f. minutissima* algae culture liquid. Bacteria adsorption rate on the algae was evaluated by preliminary experiments [31]. The adsorption rate was assessed as the ratio between the number of colonies from filtrated suspension and from total inoculum, which was calculated as 99%. Concentrations of algae and bacteria were monitored daily with 2800 UV–visible spectrophotometer (Unico, USA) for the duration of the experiment. Mantle tissues from 10 oysters per condition were individually sampled at the end of the interference phase (day 15 and day 30). No oyster died during the 15-day and 30-day exposure experiments. A total of 120 oysters were dissected. All experiments were performed in accordance with our institutional guidelines.

### 4.4. Phenotypic Change Statistical Analyses

Computer vision was used to obtain the objective and non-destructive assessment of color patterns in non-uniformly colored surfaces. *L*a*b** international standard for color measurements was used to measure the shell color changes though experiments. *L** is the lightness component, which ranges from 0 to 100, and parameters *a** (from green to red) and *b** (from blue to yellow) range from −120 to 120 [42]. The photographs of oyster shells were taken by a digital camera (Nikon D4). By comparing with the photo at the start of the RNAi experiment, individual oysters from the six groups were classified to either be having the newly deposited shell or not, and the details could be found in Table 1. Oysters which had a newly deposited shell were used to analyze the *L**, *a** and *b** values. The lightness, *a*, and *b* values obtained in Photoshop CS6 (Adobe System Incorporated, USA) were converted into *L*a*b** values using the following formulas [43]:$$L^{*}=\frac{\mathrm{Lightness}}{255}\times 100$$
$$a^{*}=\frac{240a}{255}-120$$
$$b^{*}=\frac{240b}{255}-120$$

Differences between groups (the color parameter) were investigated using a two-tailed Student’s *t* test. For all statistical results, a probability of *p* < 0.05 was considered significant. Analyses were performed using IBMSPSS Statistics 20.0 (SPSS, USA). Probabilities for two-tailed, paired *t* tests for differences in *L*a*b** value among the control and experimental groups of *C. gigas* could be found in Appendix A.

### 4.5. Total RNA Extraction and qPCR Analyses

RNA extraction was performed on the individual sample using the RNA-easy isolation reagent (Vazyme, Nanjing, China), following the manufacturer’s instruction. The total RNA of 1 μg was subjected to reverse transcription using HiScript III 1st Strand cDNA Synthesis Kit with gDNA wiper (Vazyme, Nanjing, China). Relative quantification of gene expression was estimated for each gene using the ^ΔΔ^Ct method [44]. Primer sets were designed from full cDNA sequences of *CgALAS*, *CgALAD*, *CgPBGD*, *CgUROS*, *CgUROD*, *EGFP* [26], and *elongation factor I* (*Ef1*) which were used as reference genes (Table 2). A dissociation curve was generated in each case to check that only a single band was amplified. The primer sets (*CgALAS**-Q*-*O*) for qPCR analysis of endogenous ALAS mRNA levels were derived from the region outside the cloned fragment of *CgALAS* to prevent the amplification of RNAi products (*CgALAS* dsRNA). The primer sets (*CgALAS**-Q*-*I*) were designed in the region inside the cloned fragment. Similarly, the primer sets *CgALAD**-Q-O*, *CgPBGD**-Q-O*, *CgUROS**-Q-O*, *CgUROD**-Q-O* and *CgALAD**-Q-I*, *CgPBGD**-Q-I*, *CgUROS**-Q-I*, *CgUROD**-Q-I* were designed outside and inside of the cloned fragment of *CgALAS*, *CgALAD*, *CgPBGD*, *CgUROS*, *CgUROD*, respectively. The qPCR was carried in triplicate for each sample on a LightCycler 480 real-time PCR instrument (Roche Diagnostics, Burgess Hill, UK) using ChamQ SYBR Color qPCR Master Mix (Vazyme, Nanjing, China). Cycling parameters were 95 °C for 5 min and then 40 cycles at 95 °C for 5 s and 60 °C for 20 s. Melting curve analyses were performed following the PCR to verify specific amplification. PCR efficiency (E) was assessed for each primer pair by determining the slope of standard curves obtained from cDNA serial dilution analyses of different experimental samples. Relative gene expression data were analyzed using the comparative threshold cycle (Ct) 2^−^^ΔCt^ method, where the ΔCt = Ct (interest gene)−Ct (housekeeping gene). Interest gene values were normalized in each sample with *Ef1* housekeeping gene level, as no significant differences in Ct values were observed in *Ef1* among the conditions. Results were expressed as the number of copies of gene per copy of *Ef1.* Statistical analyses were performed using the GraphPad Software 8 with an independent *t*-test. Differences were deemed statistically significant at *p* < 0.05.The detection of RNAi products (interest gene dsRNA) in oyster mantles was assessed as the mean ± SE of individuals 2^−(^^ΔCt(*CgInside*)–^^ΔCt(*CgOutside*))^, i.e., the ratio of the *CgALAS* mRNA level measured with primer sets *CgALAS-I* (quantification of both endogenously expressed *CgALAS* mRNAs and *CgALAS* dsRNAs) to the *CgALAS* mRNA level obtained with primer sets *CgALAS-O* (to quantify the endogenously expressed *CgALAS* mRNAs). Therefore, this ratio is equal to (*gene* dsRNAs + *gene* mRNAs)/*gene* mRNAs = (*gene* ds RNAs/*gene* mRNAs) + 1.

## 5. Conclusions

In conclusion, this study is further evidence of the bacteria-feeding RNAi with algae as the carrier of *E. coli* HT115 that produces dsRNA, providing an effective long-lasting method to trigger specific RNAi responses of *C. gigas*. When genes related to the accumulation of uroporphyrin (e.g., *CgALAS*, *CgPBGD*) were knocked down, the newly deposited shell of *C. gigas* showed more whiteness and less orange. Conversely, when the genes involved in uroporphyrinogen degradation were interfered (e.g., *CgUROS*) with, the newly deposited shell exhibited more orange. The study indicated that key haem pathway genes possess a vital role in porphyrin pigmentation, and thus influenced the shell color formation of *C. gigas*.

## Figures and Tables

**Figure 1 ijms-22-06120-f001:**
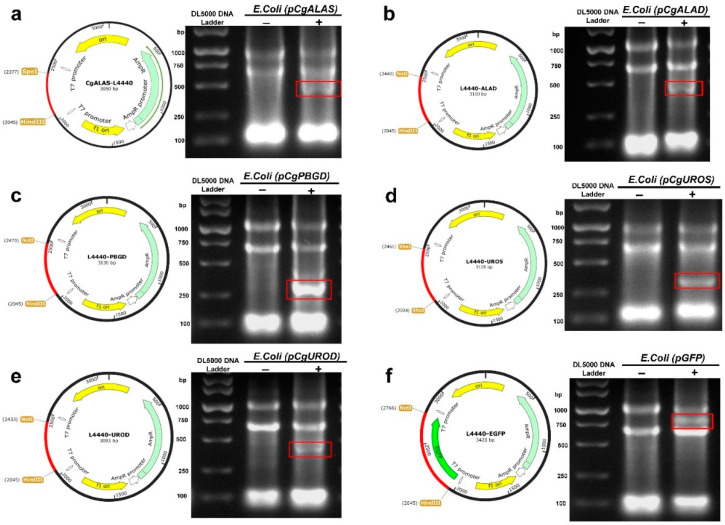
Plasmid constructs and induction assays. The red box indicates bands associated with ALAS, ALAD, PBGD, UROS, UROD and EGFP double-stranded RNAs, from the RNA extraction of the IPTG-induced *E. coli* transformed with *CgALAS*-L4440, *CgALAD*-L4440, *CgPBGD*-L4440, *CgUROS*-L4440, *CgUROD*-L4440, EGFP-L4440: (**a**) Plasmid construction of *CgALAS*-L4440 with *CgALAS* fragment sizes and total plasmid sizes; (**b**) Plasmid construction of *CgALAD*-L4440 with *CgALAD* fragment sizes and total plasmid sizes; (**c**) Plasmid construction of *CgPBGD*-L4440 with *CgPBGD* fragment sizes and total plasmid sizes; (**d**) Plasmid construction of *CgUROS*-L4440 with *CgUROS* fragment sizes and total plasmid sizes; (**e**) Plasmid construction of *CgUROD*-L4440 with *CgUROD* fragment sizes and total plasmid sizes; (**f**) Plasmid construction of EGFP-L4440 with EGFP fragment sizes and total plasmid sizes.

**Figure 2 ijms-22-06120-f002:**
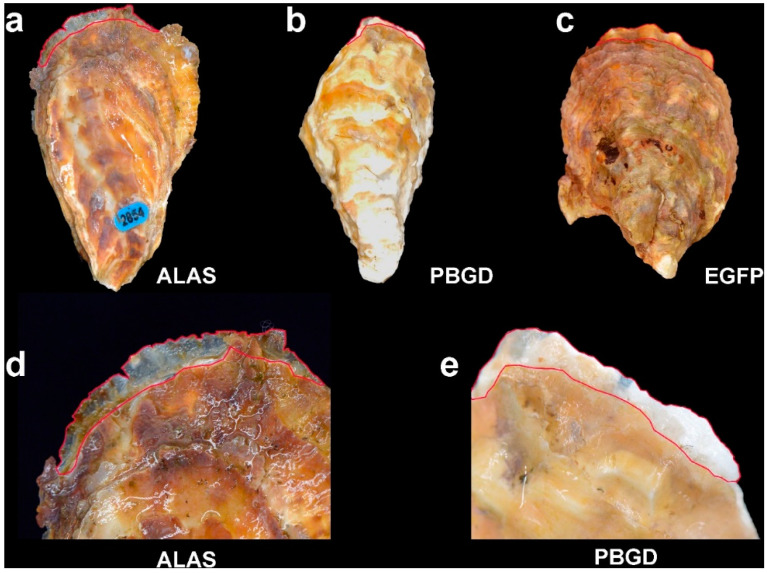
Analysis of pigmentation in newly deposited shell. Newly deposited shells are highlighted in red. (**a**) Photographic representation of *C. gigas* from group ALAS. (**b**) Photographic representation of *C. gigas* from group PBGD. (**c**) Photographic representation of *C. gigas* from group EGFP. (**d**) Bigger image for newly deposited shell *C. gigas* from group ALAS. (**e**) Bigger image for newly deposited shell *C. gigas* from group PBGD.

**Figure 3 ijms-22-06120-f003:**
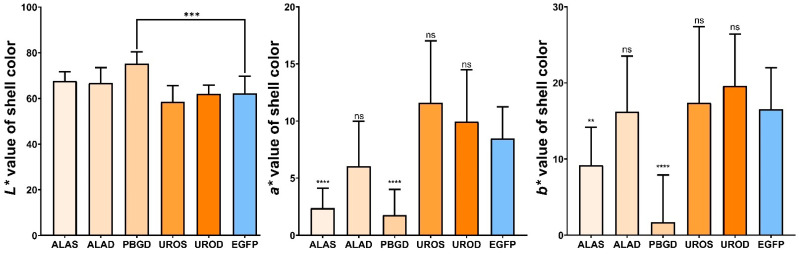
The analysis of *L*a*b** value of the newly deposited shell. Comparison of *L*a*b** value of newly deposited shell between groups *CgALAS*, *CgALAD*, *CgPBGD*, *CgUROS*, *CgUROD* and the EGFP group. The *p* values from a *t*-test demonstrate ** *p* < 0.01, *** *p* < 0.0005 or **** *p* < 0.0001; and ns demonstrates no significance.

**Figure 4 ijms-22-06120-f004:**
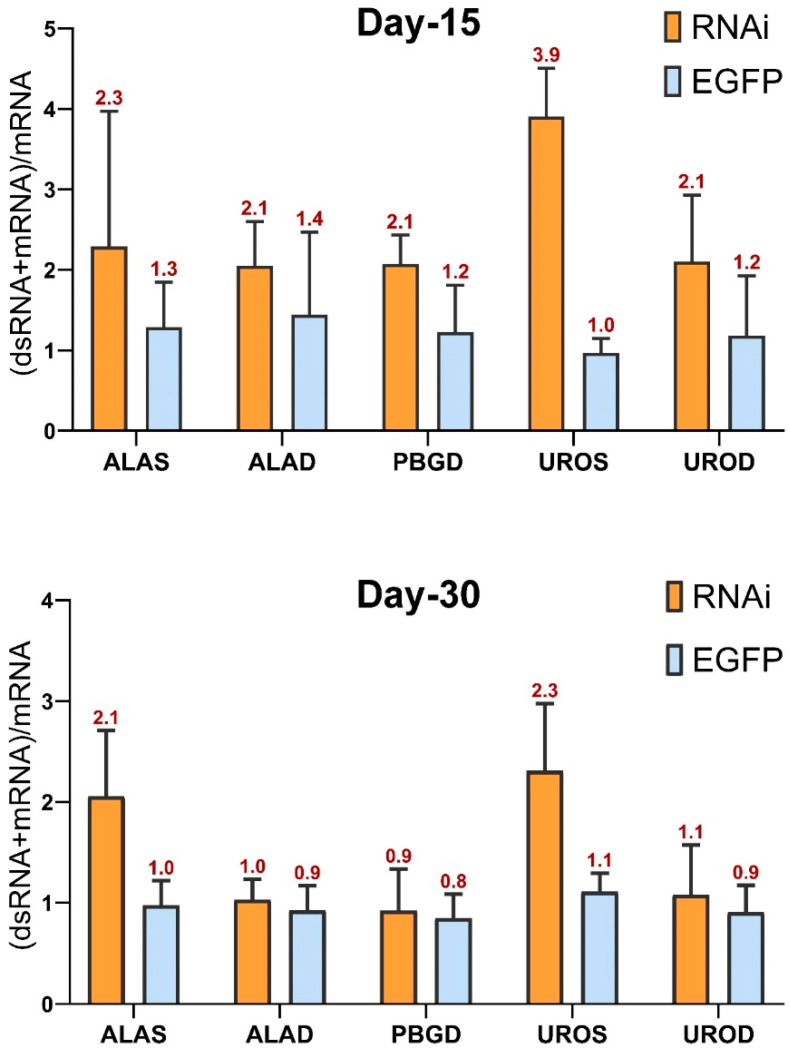
Detection of RNAi products in mantle tissues. The ratio of the interfering *CgALAS*, *CgALAD*, *CgPBGD*, *CgUROS*, *CgUROD* dsRNAs to the endogenous *CgALAS*, *CgALAD*, *CgPBGD*, *CgUROS*, *CgUROD* mRNA level are in marked red on the top of each column. The orange bars represent the ratio of *CgALAS*, *CgALAD*, *CgPBGD*, *CgUROS*, *CgUROD* of the RNAi groups, respectively, and the blue bars represent the *CgALAS*, *CgALAD*, *CgPBGD*, *CgUROS*, *CgUROD* genes ratio of the EGFP group, respectively.

**Figure 5 ijms-22-06120-f005:**
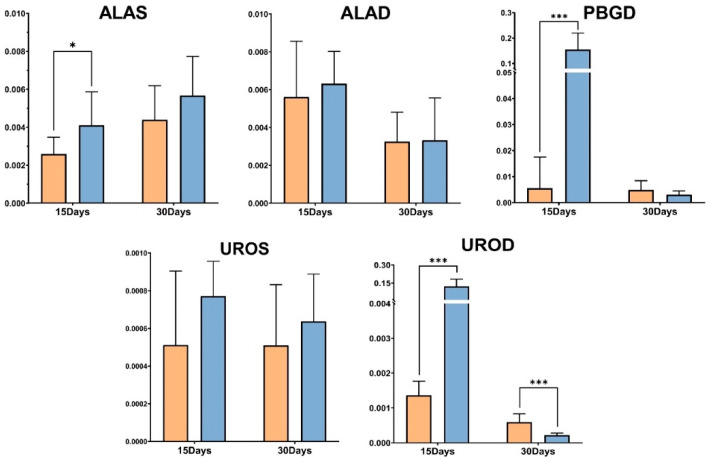
Quantitation of gene expression levels after RNAi treatment in *C. gigas*. Levels of respective gene transcripts relative to *EfI* transcripts analyzed by qPCR and expressed as the number of copies of interest gene per copy of *EfI* in the mantle of oysters at the end of interference day 15 and day 30 phases, expressed in fold change. The orange bars represent the gene expression level of the RNAi group and the blue bar represents the gene expression level of the EGFP group. The *p* values from a *t*-test demonstrate * *p* < 0.05 and *** *p* < 0.0005.

**Table 1 ijms-22-06120-t001:** Phenotypic change of *C. gigas* characterized by the whole orange shell after dsRNA interference (individual was denoted as − or + corresponding to showing the obviously newly deposited shell or not.).

Group		No. 1	No. 2	No. 3	No. 4	No. 5	No. 6	No. 7	No. 8	No. 9	No. 10
ALAS	Day 15	−	−	−	−	+	−	−	−	−	−
	Day 30	+	−	−	−	+	−	+	+	+	+
ALAD	Day 15	−	+	+	−	+	−	−	−	+	−
	Day 30	+	+	−	−	+	+	+	−	−	−
PBGD	Day 15	+	−	+	−	−	+	−	+	−	−
	Day 30	+	−	−	+	−	+	−	+	−	−
UROS	Day 15	+	+	−	−	+	−	−	−	+	+
	Day 30	−	+	+	+	−	−	+	−	−	−
UROD	Day 15	+	+	−	+	−	−	−	+	−	−
	Day 30	−	+	+	−	+	−	+	+	+	−
EGFP	Day 15	−	−	−	−	+	+	−	+	−	−
	Day 30	+	+	+	−	+	+	+	+	−	−

**Table 2 ijms-22-06120-t002:** Nucleotide sequences of specific primer pairs used in the present study.

Experiment	Primer Names	Primer Sequences	Product Length
Plasmid construction			
	ALAS-RNAi-F	CTCGAGCCCTCTCCATGAGGGGCTAG	436 bp
	ALAS-RNAi-R	CTAGAGTCCCTCTGCCCTACTCCTGC	
	ALAD-RNAi-F	AGCTTCAACTCTTCAGAGTGGGATCCATC	395 bp
	ALAD-RNAi-R	GCCGCATACCACAATGTCCATGACATGTATATG	
	PBGD-RNAi-F	AGCTTATTGATGCCTTGTCAGCTGCA	425 bp
	PBGD-RNAi-R	GCCGCCCAGTTTCTGTAGGCGCATGT	
	UROS-RNAi-F	CTCGAGAGACAGTGAGCAGGCAAAGAACC	437 bp
	UROS-RNAi-R	CTAGAGGATACATGGTACATTGGTTTAGGG	
	UROD-RNAi-F	AGCTTTTAGGGATAGGAATAGAAATTGAAGAAG	388 bp
	UROD-RNAi-R	GCCGCATTACTGTCAAACACCTGGAGCAA	
	EGFP-RNAi-F	AGCTTAGCAAGGGCGAGGAGCTG	714 bp
	EGFP-RNAi-R	GCCGCTTACTTGTACAGCTCGTCCATGCC	
Quantitative real-time PCR			
	ALAS-Q-I-F	ATGAGGGGCTAGAGGCTGAGATAG	184 bp
	ALAS-Q-I-F	ATGAGGGGCTAGAGGCTGAGATAG	
	ALAD-Q-I-F	AGGTACGGCATCAACACACTACG	213 bp
	ALAD-Q-I-F	AGGTACGGCATCAACACACTACG	
	PBGD-Q-I-F	AGACAGTCCTTATGATGTGGTAGTGATG	161 bp
	PBGD-Q-I-F	AGACAGTCCTTATGATGTGGTAGTGATG	
	UROS-Q-I-F	GAGAGACAGTGAGCAGGCAAAG	211 bp
	UROS-Q-I-F	GAGAGACAGTGAGCAGGCAAAG	
	UROD-Q-I-F	GGAATAGAAATTGAAGAAGGAAAGGGAATG	179 bp
	UROD-Q-I-R	CCAGCGAAGCCGATGAGAGG	
	ALAS-Q-O-F	CTTCATCTTCACCACCAGTCTCC	211 bp
	ALAS-Q-O-R	GCATAAGGCAGCATCACCAAC	
	ALAD-Q-O-F	GGGCGGGATACGGGAACAG	217 bp
	ALAD-Q-O-R	GCTAGACCAGGCTTAACCATCAG	
	PBGD-Q-O-F	CACAGCGAAATGGGAAAGGAGAG	182 bp
	PBGD-Q-O-R	CGTGGATACTTTGGCGGTAATAAGC	
	UROS-Q-O-F	AAAGGCTTTAGAGAGTCATGGTTACG	183 bp
	UROS-Q-O-R	AGTGGAGTGTCTGTTGAGTTGAGG	
	UROD-Q-O-F	CGACTCACGGAACAGAACATAGAAC	199 bp
	UROD-Q-O-R	TGGAATACAGCATACATGGATCAAGG	
	EGFP-Q-F	GTGCTTCAGCCGCTACCC	202 bp
	EGFP-Q-R	GATGTTGCCGTCCTCCTTG	
	EFI-F	CAAGAACGGAGATGCTGGTATGG	175 bp
	EF1-R	GGTGACTTTGCCCTGTGATGG	

## Data Availability

All the data used in this study have been provided in the main text and Appendix A.

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
