# Peer review of "RNA Interference by Ingested Dsrna-Expressing Bacteria to Study Porphyrin Pigmentation in Crassostrea gigas"

_ijms, 2021, doi:10.3390/ijms22116120_

Round 1

Reviewer 1 Report

In the presented work, Biyang Hu and coworkers have successfully induced knockdown of pigmentation genes in an oyster, Crassostrea gigas, by feeding the animals with dinoflagellate algae containing bacteria expressing dsRNA homologous to the target gene sequences. While similar approach is widely used for gene knockdown in roundworms and planaria, to my knowledge this has not been widely used for mollusks. Although the method is not novel (as correctly referenced, #26), I think the approach is elaborate and the presented study presents a nice confirmation for its applicability.

Although there are some language issues, the introduction is informative, and the presentation of the methods and results is adequate. The abstract needs to be reformulated. Eg. L17 “above-mentioned” -> aforementioned; L18-21: “This study proofed..” -> (for example) We show here that feeding the oysters with E.coli, containing dsRNA targeting pigmentation genes, is able to cause changes in the color of the newly deposited shell. For example, the RNAi knockdown of CgALAS and CgPBGD resulted in loss of uroporphyrin pigmentation from the shell due to the accumulation of the pigment in the oyster’s mantel.

Major issues:

I do not see that the knockdown of CgUROS would have any effect, contrary to what was claimed in the abstract and the results (L158, compare fig. 3). Substantiate or remove this claim.

It is not clear to me how dsRNA has been differentiated from the target mRNA in the RT-qPCR (methods, Fig. 4). If the knockdown is working (Fig. 5), I do not think that this analysis adds any relevant information, only confusing. I am also quite skeptical about the results as the measured dsRNA levels do not correlate at all with the knockdown of the targets (compare Fig. 5).

Fig. 5 (and relevant results & discussion): The UROS knockdown is obviously not working and I do not understand why the authors have been drawing conclusions about it (see above). Same applies to ALAD. Color codes for the bars are missing, but I assume they are the same as in Fig. 4?

Unless the authors are willing to trouble-shoot the non-successful RNAi experiments, such as designing new dsRNA constructs or perfecting their delivery, I would suggest to leave these out and focus on the results from the three gene knockdowns that worked.

Minor issues:

Some language editing is needed. Latin terms, such as in vivo, should be written in italics.

Author Response

Thank you for your patience and valuable advice,we have revised the manuscript accordingly. Please see the attachment.

Reviewer 2 Report

In the manuscript “RNA interference by ingested dsrna-expressing bacteria to study porphyrin pigmentation in Crassostrea gigas” the authors use E. Coli to deliver dsRNA to oysters to determine effects of different genes on pigmentation in newly deposited shells.  While the approach and findings may be of interest the manuscript lacks sufficient explanation to understand the results and I cannot see that the data supports their conclusions. I therefore cannot recommend this manuscript for publication in International Journal of Molecular Sciences.  Following are some of my major concerns:

1) In Figure 1 the dsRNA for the GFP control is more than twice the size of several of the targeting dsRNAs.  It would be more appropriate to use a control that is of similar size.

2) The way Table 2 is presented it seems that new shell deposits (or not) was determined at day 15 and 30, but how can there be new shell deposits in some columns at day 15 but not at day 30?  If these are all individual oysters then this needs to be described properly in the text. It is also not explained whether or not those oysters with no new shell deposits were excluded from all of the data analysis or just the phenotypic observations.

3) The authors talk about an L*a*b* value for change in pigmentation but then they show graphs for a* and b* value of shell color.  I do not follow at all how these values are used together or alone to show phenotypic changes in orange shell pigmentation. Furthermore, two identical graphs are shown next to each other for b* value.

4) Looking at the methods I can understand what was done in Figure 4 for the RNAi bars but the reader should not have to consult the methods to understand what is shown in the figure, this information should be provided in the results and figure legend.  What is not at all clear is what the GFP bars represent since there is no endogenous GFP mRNA to fit the equation on the axis. 

5) Figure 5 is not at all clear.  What do the yellow and blue bars represent? Why does the comparison go up in some cases and down in others and some have no change?   The correct control would be GFP dsRNA effects on the gene expression compared to the targeting dsRNA effects on gene expression but I cannot understand what the authors are trying to show in this figure and no supplementary materials were provided with the manuscript (Supp Figure S1 and Supp Table S2 for example are missing). Most concerning is that if the effects on gene expression are between the yellow and blue bars, then many of the dsRNAs did not downregulate the genes they were intended to and so no conclusions can be drawn about whether they are or are not involved in orange pigmentation.

Author Response

(The authors gave the same response as above.)

Round 2

Reviewer 1 Report

The authors have addressed by previous concerns sufficiently. Some language and text formatting still required, such as:

L18 E._coli with space and in italics

L69 Manila clam twice in the same sentence, replace the latter with “the animal” or similar.

Reviewer 2 Report

The authors have addressed my concerns in the revised manuscript and in their response to my comments.